# Constituents Isolated from the Leaves of *Glycyrrhiza uralansis* and Their Anti-Inflammatory Activities on LPS-Induced RAW264.7 Cells

**DOI:** 10.3390/molecules24101923

**Published:** 2019-05-18

**Authors:** Liyao Wang, Kaixue Zhang, Shu Han, Liu Zhang, Haiying Bai, Fang Bao, Yan Zeng, Jiyong Wang, Hong Du, Yingqian Liu, Zhigang Yang

**Affiliations:** 1School of Pharmacy, Lanzhou University, LanZhou 730000, China; lywang17@lzu.edu.cn (L.W.); zhangkx2018@lzu.edu.cn (K.Z.); zhangliu17@lzu.edu.cn (L.Z.); baihy14@lzu.edu.cn (H.B.); baof16@lzu.edu.cn (F.B.); yqliu@lzu.edu.cn (Y.L.); 2School of Chinese Materia Medica, Beijing University of Chinese Medicine, Beijing 100029, China; hs361015@163.com (S.H.); duhong@vip.163.com (H.D.); 3China National Traditional Chinese Medicine Co., Ltd., Beijing 100035, China; zyzy1221@126.com (Y.Z.); wangjiyong75@163.com (J.W.)

**Keywords:** *Glycyrrhiza uralensis*, licostilbene, licofuranol, anti-inflammatory activity

## Abstract

Licorice, the root and rhizome of *Glycyrrhiza uralansis* Fisch, is one of the most frequently used Traditional Chinese Medicines in rigorous clinical trials to remove toxins and sputum, and to relieve coughing. However, the aerial parts are not used so widely at present. It has been reported that the aerial parts have many bioactivities such as anti-microbial and anti-HIV activities. In this study, we aimed to discover the bioactive compounds from the leaves of *G. uralensis*. Four new compounds, licostilbene A-B (1–2) and licofuranol A-B (3–4), together with eight known flavonoids (5–12), were isolated and identified from the leaves of *G. uralensis*. Their structures were elucidated mainly by the interpretation of high-resolution electrospray mass spectrometry (HR-ESI-MS) and nuclear magnetic resonance (NMR) spectroscopic data. Compared with quercetin, which showed a 50% inhibitory concentration (IC_50_) value of 4.08 μg/mL, compounds 1–9 showed significant anti-inflammatory activities by inhibiting lipopolysaccharide (LPS)-induced nitric oxide (NO) production with IC_50_ values of 2.60, 2.15, 3.21, 3.25, 2.00, 3.45, 2.53, 3.13 and 3.17 μg/mL, respectively. The discovery of these active compounds is important for the prevention and treatment of inflammation.

## 1. Introduction

Licorice is a perennial herb of the genus *Glycyrrhiza*, mainly distributed in Xinjiang, Gansu, Ningxia and Inner Mongolia in China. The application of licorice has permeated many fields, including food, tobacco, cosmetics, health products and pharmaceuticals [1]. The pharmacopoeia of the People’s Republic of China contains *G. uralensis*, *G. glabra* and *G. inflata* as medicinal materials. Their roots and rhizomes are the medicinal parts, which mainly contain triterpenoids and flavonoids. To date, more than 400 compounds have been isolated from licorice [2]. It is reported that licorice shows significant efficacy in relieving coughing and asthma [3], hepatoprotective [4], anti-inflammatory [5], anti-virus [6], anti-ulcer [7] and anti-diabetes [8] activities.

The underground part of licorice has been used for the treatment of many diseases for a long time in China. The aerial parts of licorice are not widely used, usually resulting in the waste of resources. However, the aerial parts possess a high nutritional value [9]. Several compounds have been identified in the aerial parts, including 75 flavonoids, 16 dihydrostilbenes and 18 amino acids [10]. It has been reported that the aerial parts have numerous beneficial effects, such as anti-coagulation, anti-thrombosis [11], anti-oxidation [12,13], anti-prostatitis [14], anti-microbial [15] and other biological activities.

Inflammation is the body’s stress feedback to noxious stimuli, tissue damage, and infections, leading to inflammatory diseases, such as infections, metabolic diseases, cancer and aging [16]. Macrophages are important immune cells and play an important role in the inflammatory process. NO is one of important inflammatory mediators secreted by macrophages [17], and excessive production of NO will lead to cell damage and tissue necrosis, promoting the occurrence of inflammatory diseases. Thus, the inhibition of NO production is important for the treatment of inflammation and complications.

In this study, we aimed to find the bioactive compounds from the leaves of *G. uralensis*. As a result, four new compounds (**1**–**4**), together with eight known compounds (**5**–**12**), were isolated and identified (Figure 1). In addition, compounds **1**–**9** showed stronger anti-inflammatory activities by suppressing NO production in RAW264.7 cells.

## 2. Results and Discussion

### 2.1. Identification of Compounds

Compound **1** was obtained as a brown oil. The HR-ESI-MS date of **1** showed an [M + H]^+^ ion at *m*/*z* 353.1742, corresponding to a molecular formula of C_22_H_24_O_4_ (Appendix A). The UV spectrum of **1** had absorption maxima at 250, 259 and 284 nm, which indicated that **1** had an aromatic heterocyclic ring (Appendix A).

The ^1^H-NMR spectrum (Appendix A) showed four aromatic protons (Table 1). Two protons at δ_H_ 6.58 (1H, br s) and 6.75 (1H, d, *J* = 1.2 Hz) displayed meta-coupling aryl moiety and other proton resonances at δ_H_ 6.58 (1H, br s) and 6.43 (1H, d, *J* = 1.6 Hz) revealed one tetra-substituted benzene ring. Three signals at δ_H_ 3.18 (2H, d, *J* = 7.2 Hz), 5.11 (1H, t, *J* = 7.2 Hz) and 1.66 (6H, s) were due to one prenyl group. The signals at δ_H_ 6.83 (1H, d, *J* = 1.2 Hz) and 7.71 (1H, d, *J* = 1.2 Hz) were assigned to one benzofuran ring [18]. The signal of one methoxy group at δ_H_ 3.64 (3H, s) was also observed. One hydroxy was assigned to C-3 according to the down-shifted chemical signal of C-3 at δ_C_ 149.7 compared to that of **5**. In the analysis of ^13^C-NMR (Table 1), ^1^H-detected heteronuclear single quantum coherence (HSQC) and distortionless enhancement by polarization transfer (DEPT, Appendix A) spectra data revealed one methoxy at δ_C_ 59.6, one prenyl at δ_C_ 28.3, 123.4, 134.3, 25.5, 17.6, one tetra-substituted benzene ring at δ_C_ 114.5, 119.7,135.0, 149.7, 143.7, 134.3 and one benzofuran ring at δ_C_ 105.1, 143.4, 118.6, 136.9, 116.7, 155.4, 95.4, 155.38.

The prenyl group in compound **1** was linked to C-5(δ_C_ 134.3), which was confirmed by the ^1^H-detected heteronuclear multiple bond correlation (HMBC) correlations (Appendix A) of H-1′ (δ_H_ 3.18) with C-5 (δ_C_ 134.3), C-4 (δ_C_ 143.7), and C-6 (δ_C_ 119.8). The methoxy group was attached to C-4 according to the HMBC correlations of δ_H_ 3.64 (-OCH_3_) with 4-OCH_3_(δ_C_ 59.6). The connection of the benzofuran ring to C-8 was clearly proved by HMBC correlations of H-8 (δ_H_ 2.93) with C-10 (δ_C_ 118.6) and C-14 (δ_C_ 111.7), of H-12 (δ_H_ 6.75) of C-10 (δ_C_ 118.6), C-11 (δ_C_ 155.4) and C-14 (δ_C_ 111.7). Other correlations, such as H-7 (δ_H_ 2.73) with C-1 (δ_C_ 135.0), C-6 (δ_C_ 119.8) and C-8(δ_C_ 34.6), H-6 (δ_H_ 6.43) with C-2 (δ_C_ 114.5), C-4 (δ_C_ 143.7) and C-7 (δ_C_ 35.7), provided further support for the confirmation of the structure. All of the above information determined that the structure of **1** was 3-hydroxy-1-(2-(3-hydroxy-4-methoxy-5-(3-methyl-2-butenyl)-phenyl)-ethyl) benzofuran, denoted as licostilbene A.

Compound **2** was obtained as brown oil and had the molecular formula C_25_H_32_O_4_, as inferred from the HR-ESI-MS in positive mode [M + H]^+^ at *m*/*z* 397.2362 (calcd. for C_25_H_32_O_4_, 397.2373) (Appendix A). The UV spectrum of **2** had absorption maxima at 270 and 282 nm, which indicated that it had benzene rings (Appendix A).

The A ring of **2** displayed the same proton and carbon signals as those of the A ring of α,α′-dihydro-3,5,4′-trihydroxy-4,5′-diisopentenylstilbene (**5**), and the B ring of **2** was the same as the C ring of licostilbene A (**1**) (Table 1). The ^1^H-NMR spectrum (Appendix A) showed four aromatic proton resonances at δ_H_ 6.54 (1H, d, *J* = 1.6 Hz), 6.40 (1H, d, *J* = 1.6 Hz) and 6.13 (2H, s) (Table 1). The signal at δ_H_ 3.64 (3H, s) was assigned to one methoxy group. Signals at δ_H_ 1.59 (3H, s) 1.67 (6H, s), 1.68 (3H, s), 3.19 (2H, d, *J* = 7.2 Hz), 3.12 (2H, d, *J* = 7.2 Hz) and 5.17 (2H, m) were due to two prenyl groups. One signal at δ_H_ 2.57 (4H, m) was due to two methylenes. The ^13^C-NMR (Table 1), HSQC and DEPT (Appendix A) spectra of **2** displayed 25 carbon signals, including four methyls, one methoxy group, four methylenes and six methines. HMBC correlations showed correlations of H-1′ (δ_H_ 3.19) with C-4 (δ_C_ 143.6), C-5 (δ_C_ 134.3), C-6 (δ_C_ 119.6), C-2′ (δ_C_ 123.5) and C-3′ (δ_C_ 131.0), of H-7 (δ_H_ 2.57) with C-2 (δ_C_ 114.3) and C-6 (δ_C_ 119.6), of H-1′’ (δ_H_ 3.12) with C-11 (δ_C_ 155.3), C-13 (δ_C_ 155.3), C-2′’ (δ_C_ 123.9) and C-3′’ (δ_C_ 129.0), of H-2′’ (δ_H_ 5.17) with C-4′’ (δ_C_ 17.7) and C-5′’ (δ_C_ 25.5)(Appendix A). On the basis of the above inferences, the structure of **2** was identified as 7,8-dihydro-3,11,13-trihydroxy-4-methoxy-5,12-diisopentenylstilbene and named as licostilbene B.

Compound **3** was found to be brown powder. The HR-ESI-MS spectrum of **3** gave a molecular ion peak [M + H]^+^ at *m*/*z* 387.1068, consistent with a molecular formula of C_20_H_18_O_8_ (Appendix A). The configuration at C-2” was identified to be *R* based on its optical rotation value ([α]_22 *D*_ + 0.8, dissolved in methanol) [19]. The UV spectrum of **3** showed absorption maxima at 258 and 369 nm, which indicated that compound **3** had a flavonoid skeleton [20] (Appendix A).

The ^1^H-NMR spectrum (Table 2, Appendix A) of compound **3** showed the presence of four aromatic protons, with two of them at δ_H_ 6.89 (1H, d, *J* = 8.8 Hz) and 7.85 (1H, d, *J* = 8.8 Hz) assignable to a 2′,3′,4′-trihydroxy-substituted B ring of a flavone structure and two additional signals at δ_H_ 7.69 (1H, s) and 6.52 (1 H, s) assignable to the protons on C-3 and C-8 [21]. The signal at δ_H_ 9.44 (3H, br s) was due to three hydroxys connected with C-2′, C-3′ and C-4′. The signals at δ_H_ 1.15 (3H, s) and 1.14 (3H, s) were due to two methoxy groups. The proton resonance at δ_H_ 3.05 (2H, d, *J* = 8.4 Hz) and 4.74 (1H, t, *J* = 8.4 Hz) was due to an oxygenated methine and a methylene. One hydroxy (δ_H_ 12.76) was attached to C-5 (δ 154.8) because of the intramolecular hydrogen bond which increased the chemical shift. The ^13^C-NMR (Table 2), HSQC and DEPT (Appendix A) spectra showed 20 carbon signals, with 15 of them assignable to a flavone skeleton and the remaining five carbons at δ_C_ 25.9, 25.0, 25.8, 70.1 and 91.5 assignable to a modified prenyl group.

An –OCH–CH_2_– spin system was evident in the ^1^H-NMR and ^1^H-^1^H correlation spectroscopy (^1^H-^1^H COSY, Appendix A). HMBC (Appendix A) supported the correlations between the oxygenated methine and methylene protons of this spin system and the carbon resonances at δ 108.4 and δ 166.2, and thus, the presence of a furan ring was confirmed which is fused to the flavone moiety. The presence of two isolated methyl groups belonging to a hydroxyisopropyl moiety of the furan ring located next to the methine proton was evident from the peaks at δ_H_ 1.14 (3H, s) and 1.16 (3H, s) and carbon signals at δ_C_ 25.0, and 25.9 in ^1^H and ^13^C-NMR spectra. HMBC correlations within the furan ring led to the assigned structure of compound **3**. In addition, HMBC provided support for the existence of a flavoniod skeleton, such as correlations of H-3 (δ_H_ 7.69) with C-2 (δ_C_ 147.8) and C-1′ (δ_C_ 119.9), of H-6′ (δ_H_ 7.54) with C-2 (δ_C_ 147.8), of H-8 (δ_H_ 6.52) with C-6 (δ_C_ 108.4), C-7 (δ_C_ 166.2), C-9 (δ_C_ 156.1) and C-10 (δ_C_ 103.9). Thus, the structure of the new compound **3** was elucidated as (*R*)-7-(2,3,4-trihydroxyphenyl)-4-hydroxy-2-(2-hydroxypropan-2-yl)-2,3-dihydrofuro(3,2-g)chromen-5-one and named as licofuranol A.

Compound **4** was obtained as yellow powder. The HR-ESI-MS spectrum of **4** gave a molecular ion peak [M + H]^+^ at *m*/*z* 401.1221 consistent with a molecular formula of C_21_H_20_O_8_ (Appendix A). The configuration at C-2” was identified to be *S* based on its optical rotation value ([α]_22 *D*_ − 2.1, dissolved in methanol) [19]. The UV spectrum of **4** had absorption maxima at 259 and 354 nm (Appendix A), which suggested that **4** had a flavonoid skeleton.

The ^1^H-NMR spectrum (Appendix A) showed four aromatic protons (Table 2). Three proton resonances at δ_H_ 7.55 (1H, d, *J* = 1.6 Hz), δ 7.44 (1H, dd, *J* = 1.6 Hz, 8.4 Hz) and δ_H_ 6.91 (1H, d, *J* = 8.4 Hz) were observed and displayed a typical ABX system of ring B. The signals at δ_H_ 1.15 (3H, s), 1.14 (3H, s), 3.05 (2H, d, *J* = 8.4 Hz) and 4.74 (1H, t, *J* = 8.4 Hz) were due to a modified prenyl group. The signal at δ_H_ 3.78 (3 H, s) was assigned to one methoxy group. The resonance at δ_H_ 6.52 (1 H, s) was assigned to H-8 of the A ring. The ^13^C-NMR (Table 2), HSQC and DEPT spectra (Appendix A) showed 21 carbon signals, including a flavone skeleton, a modified prenyl group and one methoxy group.

The structure of compound **4** was very similar to compound **3,** except one proton attached with C-3 and one hydroxy attached with C-2′ were replaced by one methoxy group and one proton, respectively. Two hydroxys were attached to C-3′ and C-4′ according to HMBC (Appendix A) correlations of H-2′ (δ_H_ 7.55) and H-5′ (δ_H_ 6.91) with C-1′ (δ_C_ 120.5), C-3′ (δ_C_ 145.3), C-4′ (δ_C_ 148.7) and C-3′ (δ_C_ 145.3), C-4′ (δ_C_ 148.7), C-6′ (δ_C_ 120.8), respectively. The methoxy group was attached to C-3 due to the correlation of δ 3.78_H_ (-OCH_3_) with C-3 (δ_C_ 137.7). On the basis of the above analysis, the structure of **4** was (*S*)-7-(3, 4-dihydroxyphenyl)-4-hydroxy-3-methoxy-2-(2-hydroxypropan-2-yl)-2,3-dihydrofuro(3,2-g)chromen-5-one, name as licofuranol B.

The known compounds **5**–**12** were identified as α,α′-dihydro-3,5,4′-trihydroxy-4,5′-diisopentenylstilbene (**5**) [22], glycypytilbene B (**6**) [23], luteolin (**7**) [24], quercetin-3,4′-dimethyl ether (**8**) [25], calycosin (**9**) [26], scopoletin (**10**) [27], diosmetin (**11**) [28] and echinatin (**12**) [29], comparing their spectral information with published data in the literature.

### 2.2. Assessment of Compounds ***1***–***9*** on Cellular Viability.

To assess the cell viability of compounds **1**–**9**, RAW264.7 cells were exposed to different concentrations of these compounds. The control group was treated with DMSO. As a result, incubation of cells with **1**–**9** for 24 h exerted no significant influence on cell viability (Figure 2).

### 2.3. Anti-Inflammatory Activity

It has been reported that the extract of licorice suppresses LPS-induced inflammatory response in murine macrophage and increased the survival rate in LPS-induced mice macrophages [30]. In this work, the anti-inflammatory activity of compounds was evaluated in RAW264.7 cells with inflammation induced by LPS. Compounds **1**–**9** were tested for their NO inhibitory activities in the process of inflammation, and quercetin was used as a positive control. The anti-inflammatory activity of compounds **1**–**9** is shown in Table 3. As a result, four new compounds (**1**–**4**) and five known compounds (**5**–**9**) showed stronger anti-inflammatory activities than the positive control quercetin by inducing NO production on LPS-induced RAW264.7 cells.

## 3. Experimental

### 3.1. Materials and Methods

#### 3.1.1. Plant Materials

Fresh leaves of *Glycyrrhiza uralensis* were collected from Minqin County, Gansu Province, People’s Republic of China, in September 2017. The plant was identified by one of the authors, Zhigang Yang, as the voucher specimen (No. MQG201610). 

#### 3.1.2. Chemicals and Instruments

Sephadex LH-20 was purchased from GE Healthcare Bio-Sciences AB (Uppsala, Sweden). The silica gel for column chromatography and thin-layer chromatography was obtained from Qingdao Marine Chemical Company (Qingdao, China). Open column chromatography was also performed using ODS-C_18_ (Cosmosil 75 C_18_-OPN). The semipreparative HPLC system was carried out on an easysep-1050 system equipped with a single-wavelength detector which was purchased from Unimicro Technologies Co., Lid (Shanghai, China), and a Waters C18 column. ^1^H-NMR, ^13^C-NMR, and 2D-NMR spectra were obtained on a Bruker AVANCE AV III-400 spectrometer (Karlsruhe, BW, Germany). HR-ESI-MS was run on an Agilent LCxLC-IM-QTOF-MS (1290 × 1290 − 6560) spectrometer (Palo Alto, CA, USA). Optical rotations were recorded on a Perkin-Elmer 341 polarimeter. UV spectra were obtained with a UV/visible spectrophotometer (Perkin-Elmer, Waltham, MA, USA).

### 3.2. Extraction and Isolation

Leaves of *Glycyrrhiza uralensis* were dried in the shade. Extraction was performed by maceration of dried leaves (8 kg) in 80% methanol (80 L) at room temperature in a conical flask. This process was repeated 3 times and the final extract was filtered. The extract was concentrated under vacuum at 40 °C on a rotary evaporator (IKA^®^-Werke GmbH & Co.KG, Freiburg, BW, Germany). Then, all of the extract (1196.5 g) was dissolved in water and mixed with trichloromethane, ethyl acetate and n-butyl alcohol in a separating funnel to obtain four layers. The ethyl acetate layer was subjected to silica gel (100–200 mesh) column chromatography by successive elution with different proportions of petroleum ether/ethyl acetate (100:0–0:100). The extract was separated into 9 fractions (fr.A-I) according to TLC spots.

Fr.G (10.88 g) was separated by a reverse phase silica gel column with different proportions of methanol/water (20:80–100:1) to obtain 8 fractions (fr.G1-G8). Fr. G2 was further purified by semipreparative HPLC (MeOH/H_2_O, 70:30) to obtain scopoletin (**10**, 2.0 mg). Fr.G4 and Fr.G5 were subjected to silica gel column chromatography to obtain fr.G4A-H and fr.G5A-G, respectively. Fr.G4E was further purified by semipreparative HPLC (MeOH/H_2_O 53:47) to obtain calycosin (**9**, 6.7 mg) and echinatin (**12**, 4.6 mg).

Luteolin (**7**, 5.1 mg) was obtained from fr.G4F by semipreparative HPLC (MeOH/H_2_O 58:42). Fr.G5C was also further purified by semipreparative HPLC (MeOH/H_2_O 58:42) to obtain diosmetin (**11**, 7.9 mg) and quercetin-3,4′-dimethyl ether (**8**, 4.2 mg). Fr.G5E was applied to the silica gel column and eluted with CH_2_Cl_2_/MeOH to obtain 4 fractions (fr.G5E1-4). The semipreparative HPLC of fr.G5E2 (MeOH/H_2_O 60:40) led to licofuranol A (**3**, 26.5 mg) and licofuranol B (**4**, 15.8 mg).

Fraction D (1.85 g) was fractionated on a silica gel column to obtain 6 fractions (fr. D1-6). fr.D5 was separated by silica gel chromatography and successively eluted with petroleum ether/acetone(100:0-0:100) to obtain 8 fractions (sub-fr.D5A–H). D5G was further purified by silica gel column chromatography (MeOH/H_2_O 75:25) to give four compounds including licostilbene A (**1,** 18.5 mg), licostilbene B (**2**, 8.9 mg), α,α′-dihydro-3,5,4′-trihydroxy-4,5′-diisopentenylstilbene (**5,** 12.6 mg) and glycypytilbene B (**6**, 28.4 mg).

### 3.3. Anti-Inflammatory Activity

#### 3.3.1. Cell Culture

The RAW264.7 macrophage cell line was obtained from the National Infrastructure of Cell Line Resource, maintained in Dulbecco’s modified Eagle medium (DMEM, 10% fetal bovine serum, 1% penicillin and streptomycin) and grown at 37 °C in a humidified incubator containing 5% CO_2_. DMEM was changed every two days and cells were subcultured when 80–90% confluency was reached.

#### 3.3.2. MTT Assay

The viability of RAW264.7 cells in the presence of different concentrations of compounds (**1**, **3**, **5** μg/mL, dissolved in DMSO) was evaluated by MTT assay. Briefly, the cells (1 × 10^5^ cells/mL) were seeded into a 96-well plate. After 12 h incubation, compounds **1**–**9** with different concentrations were added into the wells and incubated for 24 h. Following incubation, the medium was removed, and 20 μL of MTT (5 mg/mL) was added into the wells and incubated for another 4 h. Finally, 150 μL of DMSO was added and the absorbance at 490 nm was determined using a microplate reader.

#### 3.3.3. NO Determination

RAW264.7 cells were seeded at 1 × 10^6^/mL in 96-well culture plates for NO. Nitrite (NO_2_^−^) in the culture medium was measured as an indicator of NO production using the Griess reaction. All of the operation was according to the NO kit instructions. Briefly, the cells were separated into four groups: treated with DMSO (negative control), LPS (1 μg/mL, model), LPS + quercetin (5 μg/mL, positive control) and LPS + compounds with different concentrations (treatment group). All the agents were added at the same time, and the groups were treated for 24 h. After treatment with various concentrations of compounds and quercetin, the supernatant of cells was mixed with an equal volume of Griess reagent, and the absorbance of the mixture was measured at 540 nm. The experiments were repeated three times independently.

## 4. Conclusions

In this work, 12 compounds, including four new structures (**1**–**4**), were isolated and identified from leaves of *G. uralensis*. Among these compounds, **1**–**9** showed significant NO inhibitory activities on LPS-induced RAW264.7 cells. Four bioactive compounds, licostilbene A–B (**1**–**2**) and licofuranol A–B (**3**–**4**), were observed for the first time. We hope that these active compounds will be useful for the prevention and treatment of inflammation.

## Figures and Tables

**Figure 1 molecules-24-01923-f001:**
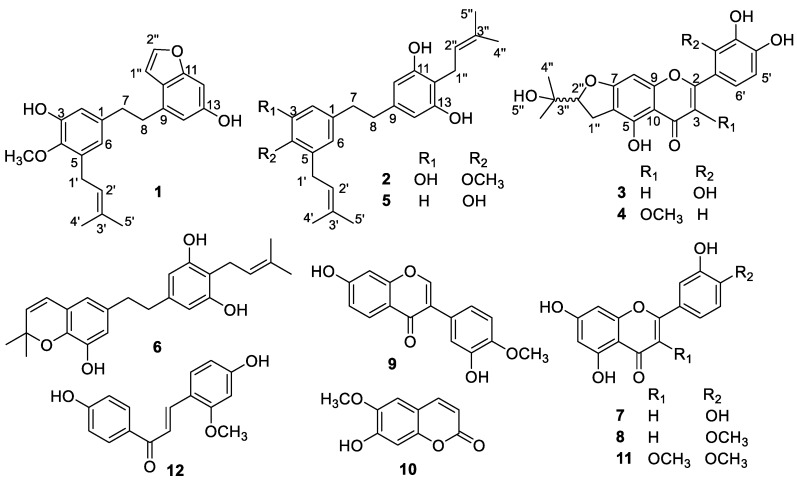
Structures of compounds **1**–**12** from *G. uralensis.*

**Figure 2 molecules-24-01923-f002:**
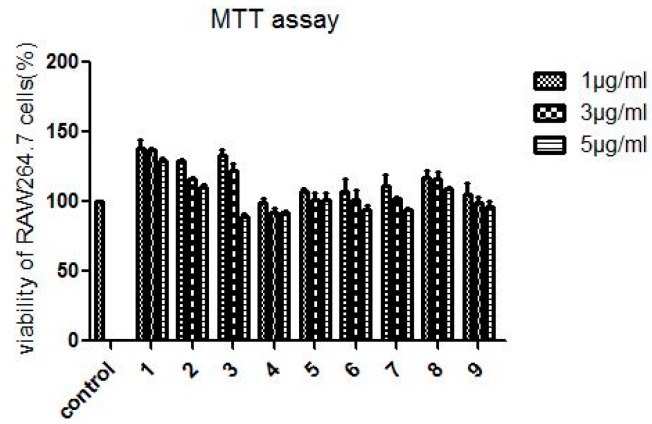
The cellular viabilities of RAW264.7 cells treated with compounds **1**–**9**. All the data are expressed as means ± SD of three independent experiments.

**Table 1 molecules-24-01923-t001:** ^1^H- and ^13^C-NMR spectroscopic data of compounds **1**, **2** and **5** (400 MHz for ^1^H-NMR and 100 MHz for ^13^C-NMR, in DMSO-*d*_6_, *δ* in ppm, *J* in Hz).

Position	1		2		5	
	δ_H_	δ_C_	δ_H_	δ_C_	δ_H_	δ_C_
1	-	135.0	-	137.1	-	131.8
2	6.58 (1H, s)	114.5	6.54 (1H, d, *J* = 1.6 Hz)	114.3	6.82 (1H, d, *J* = 2 Hz)	129.2
3	-	149.7	-	149.7	-	127.1
4	-	143.7	-	143.6	-	152.9
5	-	134.3	-	134.3	6.67 (1H, d, *J* = 8.8 Hz)	114.7
6	6.43 (1H, d, *J* = 1.6 Hz)	119.8	6.40 (1H, d, *J* = 1.6 Hz)	119.6	6.80 (1H, dd, *J* = 8.8, 2 Hz)	126.2
7	2.73 (2H, m)	35.7	2.57 (2H, m)	36.5	2.62 (2H, m)	36.4
8	2.93 (2H, m)	34.6	2.57 (2H, m)	37.1	2.54 (2H, m)	37.6
9	--	136.9	-	139.6	-	139.7
10	-	118.6	6.13 (1H, s)	106.2	6.12 (1H, s)	106.3
11	-	155.4	-	155.7	-	155.7
12	6.75 (1H, d, *J* = 0.8 Hz)	95.4	-	111.7	-	111.7
13	-	155.4	-	155.7	-	155.7
14	6.58 (1H, s)	111.7	6.13 (1H, s)	106.2	6.12 (1H, s)	106.3
1′	3.18 (2H, d, *J* = 7.2 Hz)	28.3	3.19 (2H, d, *J* = 7.2 Hz)	28.3	3.16 (2H, d, *J* = 7.2 Hz)	28.2
2′	5.16 (1H, t, *J* = 7.2 Hz)	123.5	5.17 (1H, m)	123.5	5.24 (1H, t, *J* = 7.2Hz)	123.1
3′	-	130.9	-	131.0	-	130.9
4′	1.66 (3H, s)	17.7	1.67 (3H, s)	17.7	1.65 (3H, s)	17.7
5′	1.66 (3H, s)	25.5	1.67 (3H, s)	25.5	1.67 (3H, s)	25.5
1′′	6.83 (1H, d, *J* = 2.0 Hz)	105.1	3.12 (2H, d, *J* = 6.8 Hz)	21.9	3.12 (2H, d, *J* = 6.8 Hz)	21.9
2′′	7.71 (1H, d, *J* = 2.0 Hz)	143.4	5.17 (1H, m)	123.9	5.15 (1H, t, *J* = 6.8 Hz)	123.9
3′′	-	-	-	129.0	-	129.0
4′′	-	-	1.59 (3H, s)	17.7	1.59 (3H, s)	17.7
5′′	-	-	1.68 (3H, s)	25.5	1.68 (3H, s)	25.5
4-OCH_3_	3.64 (3H, s)	59.6	3.64 (3H, s)	59.6		

**Table 2 molecules-24-01923-t002:** ^1^H- and ^13^C-NMR spectroscopic data of compounds **3**–**4** (400 MHz for ^1^H-NMR and 100 MHz for ^13^C-NMR, in DMSO-*d*_6_, *δ* in ppm, *J* in Hz)**.**

Position	3		4	
	δ_H_	δ_C_	δ_H_	δ_C_
2	-	147.8	-	155.6
3	7.69 (1H, s)	115.6	-	137.7
4	-	176.0	-	178.1
5	-	154.8	-	155.3
6	-	108.4	-	108.8
7	-	166.2	-	166.3
8	6.52 (1H, s)	88.4	6.52 (1H, s)	88.6
9	-	156.1	-	156.3
10	-	103.9	-	105.1
1′	-	119.9	-	120.5
2′	-	146.9	7.55 (1H, d, *J* = 1.6 Hz)	115.5
3′	-	135.8	-	145.3
4′	-	145.1	-	148.8
5′	6.88 (1H, d, *J* = 8.8 Hz)	115.2	6.91 (1H, d, *J* = 8.4 Hz)	115.7
6′	7.54 (1H, d, *J* = 8.8 Hz)	122.0	7.44 (1H, dd, *J* = 1.6, 8.4 Hz)	120.8
1′′	3.06 (2H, d, *J* = 8.4 Hz)	25.8	3.05 (2H, d, *J* = 8.4 Hz)	25.8
2′′	4.73 (1H, t, *J* = 8. 4Hz)	91.5	4.74 (1H, t, *J* = 8.4 Hz)	91.6
3′′	-	70.1	-	70.1
4′′,5′′	1.16, 1.14 (6H, d)	25.0,25.9	1.15, 1.14 (6H, d)	24.9,25.9
3-OCH3	-	-	3.78 (3H, s)	59.7
5-OH	12.76 (1H, s)	-	12.92 (1H, s)	-

**Table 3 molecules-24-01923-t003:** Effect of compounds **1**–**9** on NO production stimulated by LPS in RAW264.7 cells.

No.	IC_50_ (μg/mL)
**1**	2.60
**2**	2.15
**3**	3.21
**4**	3.25
**5**	2.00
**6**	3.45
**7**	2.53
**8**	3.13
**9**	3.17
quercetin	4.08

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
