# Peer review of "Constituents Isolated from the Leaves of Glycyrrhiza uralansis and Their Anti-Inflammatory Activities on LPS-Induced RAW264.7 Cells"

_molecules, 2019, doi:10.3390/molecules24101923_

Round 1

Reviewer 1 Report

The authors should consider the following suggestions:

1.           Raw 264.7 cell viability test of those concentrations of the compound(s) should be performed. Assays such as MTT assay should be performed to make sure the reduction of nitric oxide release was not due to the reduction of cells present (killed by higher dosage of the compound)

2.           Please mentioned whether the intrinsic color (or OD) of the compound interfere with the readings of the Griess assay. If yes, did the author subtract such intrinsic OD with the blank?

3.           The type of LPS as well as their dose should be mentioned in the manuscript.

4.           The authors are advised to make a table, listing the % composition of those compounds (1 to 12) present in the leaves of Glycyrrhiza uralansis.

5.           Please state clearly the novelty of this research in your abstract and conclusion.

6.           The authors should consider using English proof-reading services by language professional.

7.           Please declare whether the content of the manuscript may have any financial affiliation with the author, from “China National Traditional Chinese Medicine Co.,Ltd”.

Author Response

Response to Reviewer 1 Comments

Reviewer #1:

Comment 1: Raw 264.7 cell viability test of those concentrations of the compound(s) should be performed. Assays such as MTT assay should be performed to make sure the reduction of nitric oxide release was not due to the reduction of cells present (killed by higher dosage of the compound)

Response: We conducted MTT assay and added relevant contents in the manuscript.

Comment 2: Please mentioned whether the intrinsic color (or OD) of the compound interfere with the readings of the Griess assay. If yes, did the author subtract such intrinsic OD with the blank?

Response: Thank you very much, in our study, the intrinsic colors of these compounds had no influence on OD values in the Griess assay at 540 nm.

Comment 3: The type of LPS as well as their dose should be mentioned in the manuscript.

Response: LPS was purchased from Beijing lambold biotechnology co. LTD, and the dose (1μg/ml) of LPS was added into the manuscript.

Comment 4: The authors are advised to make a table, listing the % composition of those compounds (1 to 12) present in the leaves of Glycyrrhiza uralensis.

Response: We have added the table of % composition into supplementary material.

Comment 5: Please state clearly the novelty of this research in your abstract and conclusion.

Response: We have stated the novelty of this research in the abstract and conclusion.

Comment 6: The authors should consider using English proof-reading services by language professional.

Response: We have corrected the manuscripts by using a professional English editing service from MDPI.

Comment 7: Please declare whether the content of the manuscript may have any financial affiliation with the author, from “China National Traditional Chinese Medicine Co.,Ltd”.

Response: Our research was partly supported by the projects of 2017YFC1701400 and SQ2017YFC170424, which are hosted by “China National Traditional Chinese Medicine Co., Ltd”.

Reviewer 2 Report

In this present study investigated “Constituents isolated from the leaves of Glycyrrhiza  uralansis and their anti-inflammatory activities on LPS-induced RAW264.7 cells”. There are some concerns to be addressed:

1. Reference format is not suitable.

2.Abstract is too rough. The authors should edit Abstract.

3. Materials and Methods: in 3.3.2. NO determination, How dose LPS be used in culture medium.

4. the experiment anti-inflammatory activity is too rough. The culture medium could assay proinflammatory cytokines, PGE2. The cell protein could detect iNOS, COX-2,

MAPK and NF-kB signal.

5.Table 3 is about anti-inflammatory activity of compounds 1-9. But, no any information describe how to detect and measure the NO levels.

6. compounds 1-9 is new compound or some compounds had been confirmed by other authors.

7. Please clearly formulate the conclusions.

Author Response

Response to Reviewer 2 Comments

Reviewer #2:

Comment 1: Reference format is not suitable.

Response: We have made correction of references.

Comment 2: Abstract is too rough. The authors should edit Abstract.

Response: Thanks for your suggestion, we have re-written the abstract.

Comment 3: Materials and Methods: in 3.3.2. NO determination, how dose LPS be used in culture medium.

Response: The dose of LPS is 1μg/ml. We have added it in the manuscript.

Comment 4: the experiment anti-inflammatory activity is too rough. The culture medium could assay proinflammatory cytokines, PGE2. The cell protein could detect iNOS, COX-2, MAPK and NF-kB signal.

Response: We mainly focused on the isolation and identification of compounds and their NO inhibitory activities in our experiment. It is difficult for us to detect other proinflammatory cytokines under the present financial situation.

Comment 5: Table 3 is about anti-inflammatory activity of compounds 1-9. But no any information describe how to detect and measure the NO levels.

Response: We have edited the title of table 3. The detail detection and measurement methods of NO levels are in the parts of 3.3.2. NO determination, Materials and Methods.

Comment 6: compounds 1-9 is new compound or some compounds had been confirmed by other authors.

Response: We have state it in the abstract, compounds 1-4 are new compounds isolated from leaves of G. uralensis, and compounds 5-9 are known compounds confirmed by other authors before.

Comment 7: Please clearly formulate the conclusions.

Response: Thank you very much, we have edited the conclusions in the manuscript.

Round 2

Reviewer 2 Report

accept